# Ethylene Plays a Dual Role during Infection by *Plasmodiophora brassicae* of *Arabidopsis thaliana*

**DOI:** 10.3390/genes13081299

**Published:** 2022-07-22

**Authors:** Kai Wang, Yiji Shi, Qingbin Sun, Mingjiao Lu, Lin Zheng, Bakirov Aldiyar, Chengyu Yu, Fengqun Yu, Aixia Xu, Zhen Huang

**Affiliations:** 1State Key Laboratory of Crop Stress Biology for Arid Areas, College of Agronomy, Northwest A&F University, Yangling 712100, China; wangkai317@nwafu.edu.cn (K.W.); shiyj0718@nwafu.edu.cn (Y.S.); 15765587319@163.com (Q.S.); 13002759528@163.com (M.L.); zl0912@nwafu.edu.cn (L.Z.); aldiyar@nwafu.edu.cn (B.A.); yu1009@nwsuaf.edu.cn (C.Y.); xuaixia2013@163.com (A.X.); 2Saskatoon Research and Development Centre, Agriculture and Agri-Food Canada, 107 Science Place, Saskatoon, SK S7N OX2, Canada; fengqun.yu@canada.ca

**Keywords:** *Plasmodiophora brassicae*, ethylene, WRKY75, *Arabidopsis thaliana*, EIN3/EIL1

## Abstract

*Plasmodiophora brassicae* infection leads to hypertrophy of host roots and subsequent formation of galls, causing huge economic losses to agricultural producers of *Cruciferae* plants. Ethylene (ET) has been reported to play a vital role against necrotrophic pathogens in the classic immunity system. More clues suggested that the defense to pathogens in roots may be different from the acrial. The ET pathway may play a positive role in the infection of *P*. *brassicae*, as shown by recent transcriptome profiling. However, the molecular basis of ET remains poorly understood. In this study, we investigated the potential role of ethylene against *P*. *brassicae* infection in an *ein3/eil1* double-mutant of *Arabidopsis thaliana* (*A. thaliana*). After infection, *ein3/eil1* (Disease Index/DI: 93) showed more susceptibility compared with wild type (DI: 75). Then, we inoculated *A. thaliana Columbia-0* (*Col-0*) with *P*. *brassicae* by 1-aminocyclopropane-1-carboxylic acid (ACC) and pyrazinamide (PZA), respectively. It was found that the symptoms of infected roots with ACC were more serious than those with PZA at 20 dpi (day post infection). However, the DI were almost the same in different treatments at 30 dpi. *WRKY75* can be directly regulated by ET and was upregulated at 7 dpi with ACC, as shown by qRT-PCR. The *wrky75-c* mutant of *A. thaliana* (DI: 93.75) was more susceptible than the wild type in *Arabidopsis*. Thus, our work reveals the dual roles of ET in infection of *P. brassicae* and provides evidence of ET in root defense against pathogens.

## 1. Introduction

Clubroot is caused by the infection of a soilborne obligate biotrophic protist *Plasmodiophora brassicae* Woron, which is one of the most destructive diseases of *Cruciferous* crops [1], with billions of dollars lost. Clubroot is widely spreading in Brazil, South Africa, Australia, New Zealand, China, and Russia [2]. The model plant *Arabidopsis thaliana* and 16 *Brassicaceae* crops (about 3.2–4.0 million ha), including *Brassica rapa pekinensis*, *Brassica napus*, and *Brassica juncea*, are the hosts of *P*. *brassicae* [3]. The infection process of *P*. *brassicae* can be divided into two stages. Under proper conditions, primary zoospores, delivered by resting spores, penetrate root hairs and epidermal cells, and subsequently, secondary zoospores, which are produced in infected root hairs and epidermal cells, are released into the soil, This stage is known as the primary infection. The secondary infection stage occurs in cortical cells through the released zoospores, resulting in gall formation [4]. These galls are sucrose sinks that hijack the sugar partitioning of host plants via promoting phloem differentiation and phloem-specific expression of sugar transporters [5], thus blocking the uptake of nutrition and water in susceptible plants.

Plants have developed sophisticated immune systems against diverse pathogens that may affect the yield and quality of crops. In recent years, two-layered innate immune systems are widely accepted. One is triggered by the recognition of pathogen/damage-associated molecular patterns (P/DAMPs). The other one is triggered by recognition of the pathogen effectors via receptors known as surface pattern-recognition receptors (PRRs) and intracellular nucleotide-binding domain leucine-rich repeat containing receptors (NLRs). The recognition of two receptors results in pattern-triggered immunity (PTI) and effector-triggered immunity, respectively [6]. PTI and ETI share some overlapping downstream outputs, and the phytohormone signaling pathway plays a vital role in the plant immune signaling pathway network. Among them, salicylic acid (SA) and jasmonate (JA)/ethylene (ET) are the critical major defense players against biotrophic pathogens and necrotrophic pathogens, respectively [7]. However, our understanding of plant defense is mostly from the aerial part of the plant. Recently, considerable progress has been made in understanding the defense mechanisms of the root systems, and some differences have been highlighted between roots and shoots in response to pathogens [8]. Of note, both SA and JA pathways have a positive effect against the biotrophic root pathogen *P. brassicae* [9]. Although transcriptome analysis has indicated the ethylene signaling pathway may be involved in resistance to *P*. *brassicae,* experimental studies have yet to be conducted to support this hypothesis.

Ethylene, a gas phytohormone, has been shown to play a significant role in plant development and response to abiotic and biotic stresses, including salinity stress, heat stress, heavy metal stress, flooding stress, etc. [10]. Ethylene is produced from L-methionine, and the last step of the synthesis is catalyzed by ACC oxidases (ACO) with 1-aminocyclopropane-1-carboxylic acid (ACC) as the substrate. *EIN3/EIL1* transcription factors, targeted by *EIN2*, are the key regulators of ethylene signaling pathway that maintain multiple responses to ethylene [11]. Many transcription factors, such as *AP2/ERF*, *WRKY*, and *NAC*, have been shown to be regulated by ethylene through *EIN3/EIL1*. *WRKY75*, involved in various biological processes including leaf senescence, phosphate deficiency, oxalic acid stress, flowering time, defense responses, and root hair development, can be directly targeted by *EIN3* [12,13,14,15,16].

Ethylene may participate in inhibiting or stimulating cell division and expansion in different organs. For example, ethylene can promote cell division through several downstream *AP2*/*ERF* transcription factors in vascular tissue [17], yet it inhibits cell division by multiple mechanisms in leaves that are exposed to different environmental stresses [18]. In addition, ethylene exhibits positive and negative effects on cell expansion in petioles and leaves, respectively [19,20]. The RALFs–LRXs–FER pathway is thought to regulate plant immune and cell expansion. FERONIA (FER), a member of receptor-like kinases, has been found to participate in a variety of cellular processes, such as ethylene responses in Arabidopsis hypocotyls [21] and function as the receptor of RALF peptide in plant immune system [22]. RALFs and LRXs are involved in plant growth via cell wall signal transduction [23]. Rapid alkalinization factor (RALF) peptides have been reported to be involved in various processes, such as plant immune response [24], through regulating cell expansion in most cases.

Based on previous studies, it requires a deeper understanding of ET in the interaction between plant and *P*. *brassicae*. In this study, we firstly confirmed the positive role of ET by the inoculation process in the *ein3/eil1* double mutant. To further investigate ET-mediated responses in different stages during clubroot infection, exogenous ACC and ACC oxidase inhibitors were applied during *P*. *brassicae* infection in *Arabidopsis*. The altered *P*. *brassicae* symptoms and transcriptional regulation were investigated to decipher the sophisticated mechanism of ET. Our study provides a more detailed view of ET function in the interaction with *P*. *brassicae*.

## 2. Materials and Methods

### 2.1. Plant Materials and Growth Conditions

*A. thaliana Columbia-0* (*Col-0*) and three mutants, *ein3* and *eil1* single mutants and *ein3/eil1* double mutant, were provided by Dr Guo (Southern University of Science and Technology). The *wrky75-c* mutant was obtained using the CRISPR/Cas9 genome editing system in the *Arabidopsis Col-0* background through the floral dip method mediated by *Agrobacterium*.

Dried seeds were surface-sterilized with 70% ethyl alcohol for 3 min, stratified in 1/2 MS (Murashige and Skoog) medium at 4 °C for 3 d, and then placed in the growth chamber at 22 °C under 16 h light/8 h dark for 1 week. The healthy seedlings with a similar growth rate were selected to transfer into the soil (mix matrix soil, vermiculite, and perlite with 3:1:1) and grow at the same conditions in the growth chamber.

### 2.2. Vector Construction

To obtain the *wrky75-c* mutant lines through the CRISPR/Cas9 system, two sgRNAs (19 bp) in the *WRKY75* genome, G1: 5′-GGAGTCGTCGAAAAAGAAG-3′ and G2: 5′-TTAACAGTGGACCAAGAAG-3′, were designed by CRISPR-P 2.0 (http://crispr.hzau.edu.cn/CRISPR2/ (accessed on 6 July 2020)) [25]. Then, the sgRNA’s fragment was cloned with the template pCBCT_1_T_2_ and the primes containing homologous arms near BSAI F: 5′-TCGAAGTAGTGATTGAGCGACAGAGGTAACCCAAGTTTTAGAGCTAGAAATAGC-3′ and R: 5′-TTCTAGCTCTAAAACTGGTGCTTGCTGGGCTTATCAATCTCTTAGTCGACTCTAC-3′. Finally, the CRISPR/Cas9 expression vector AtU6-G1-Ter1-AtU9-G2-Ter2 (U6:G1-U9:G2) was obtained via homologous recombination connecting the sgRNA’s fragment and PBSE401, and cut by the restriction enzyme BSAI. A 35S:DsRED expression cassette was transferred into U6:G1-U9:G2 through restriction-ligase reaction via EcoRI to simplify the screening of positive mutant lines.

### 2.3. RNA Isolation and Quantitative PCR Analyses

Total RNA was isolated from the treated roots with a RNAprep Pure plant kit (TIANGEN), and the reverse transcription of RNA was performed using a Fastking RT kit (with gDNase, TIANGEN). For each RNA sample, the concentration was measured with a NanoDrop 2000 spectrophotometer, and the integrity of RNA was confirmed using RNase-free agarose gel electrophoresis. Quantitative RT-PCR was performed on the ABI QuantStudio 7 Flex Real-Time PCR system using 2× realstar green fast mixture with ROX Ⅱ (SenStar). The PCR cycling reaction was performed according to the following program: 95 °C for 10 min, followed by 40 cycles at 95 °C for 15 s, and 60 °C for 1 min. The relative gene expression was calculated using the 2^–ΔΔCt^ method [26]. Three biological replicates were carried out for each experiment, and the results were normalized using the reference gene *ACTIN2* and *UBQ10*. The amounts of RNA and cDNA used were 1μg and 200 ng, respectively. Data were analyzed in Excel Office. Significant differences in the expression levels were evaluated with the *t*-test (* *p* < 0.05; ** *p* < 0.01). All of the qRT-PCR primers are listed in Appendix A, and the most qRT-PCR primers in *Arabidopsis* were obtained at qPrimerDB (https://biodb.swu.edu.cn/qprimerdb/ (accessed on 23 March 2021)) [27]. To further understand the ET-mediated pathway in response to *P. brassicae* infection, a number of genes were detected, including SA and JA marker genes (*PR2*, *THI2* and *ARGAH2*), some transcription factors (*WRKY75*, *WRKY45*, SHN1, *ERF105* and *At2G20350*), and members of the RALFs–LRXs–FER pathway.

### 2.4. Treatment in the Inoculation Process

Healthy and consistent two-week-old seedlings were inoculated with 2 mL 1 × 10^7^ spores/mL of *P*. *brassicae pathotype 4* (P4) of China. The controls were treated with H_2_O in the same stage. For the ACC and PZA treatments, ACC and PZA were added into clubroot fluid, and the final concentrations were 100 mM, 500 mM, and 1000 mM for ACC and 50 mM, 250 mM, and 500 mM for PZA, and the inoculation process was the same as the others. After that, the phenotype of roots was photographed at 20 dpi and 30 dpi. The disease index was calculated using the following 0–4 scoring system [28] with DI = 100 × (1N1 + 2N2 + 3N3 + 4N4)/4Nt. Each treatment or lines contained at least 40 plants, and the data were analyzed in Excel 2019 using *t*-test (* *p* < 0.05; ** *p* < 0.01).

### 2.5. Mutant Identification

*A. thaliana* mutants *ein3*, *eil1*, and *ein3/eil1* were confirmed by sequencing the products of PCR. The *wrky75-c* mutant positive lines were screened by fluorescent tag DsRed and confirmed by sequencing the target region (all of the primers are listed in Appendix A).

## 3. Results

### 3.1. Phenotype Analysis of Mutants in Ethylene Signaling Pathway

To explore the potential role of ethylene in plant responses to *P*. *brassicae, Col-0, ein3/eil1* double mutant, and *ein3* and *eil1* single mutants were planted and inoculated with P4 of ten-day-old seedlings. The symptoms of the roots were observed in treated and untreated plants after 30 d (Figure 1). Compared to *Col-0* (DI = 75), *ein3/eil1* (DI = 93) was more susceptible, as indicated by the bigger galls and few or no lateral roots (Figure 2). However, *ein3* and *eil1* single mutants showed smaller galls compared with *ein3/eil1*, suggesting the redundancy function between *EIN3* and *EIL1* in the ET pathway. Ethylene may be needed in resistance to *P*. *brassicae* in *Arabidopsis*, but the details remain to be explored.

### 3.2. Exogenous Application of ACC Revealed a Dual Role during the Infection of P. brassicae

Previous studies have shown that exogenous application of plant immune hormones, including SA and JA, reduces clubroot symptoms in *Col-0* [9]. To confirm the function of ethylene, the same strategy was adopted to evaluate the role of ethylene’s response to clubroot. The clubroot symptoms were identified at 20 dpi and 30 dpi. In general, the treatment with ACC decreased the clubroot symptoms compared with PZA at 20 dpi (Figure 2), and the treatment with PZA led to bigger galls in the lateral root, especially in 250 mM PZA (Figure 3). In addition, the effect of ET and its inhibitor may be dose-dependent when no more than 500 mM ACC and 250 mM PZA were applied separately (Figure 2), which may delay or accelerate gall formation with different patterns between ACC and PZA under high concentrations. Although inoculation with 1000 mM ACC proved more severe than in those with 100 mM and 500 mM ACC, it was better than those treated with PZA. In the case of 500 mM PZA, it showed similar symptoms with those inoculated by 250 mM PZA. However, the treatment with ACC and PZA displayed similar results at 30 dpi, while the clubroots treated with ACC exhibited similar DI to those with PZA (Figure 2). Of note, the galls of the plants treated with 250 mM and 50 mM PZA displayed slight root at 30 dpi, which is consistent with previous results that ethylene delayed the invasion of *P. brassicae* (Figure 4). The development of galls may be induced by ethylene from 20 dpi to 30 dpi. Taken together, we conclude that exogenous ethylene may delay the infection of *P. brassicae.* However, once the invasion is established, ethylene may promote the development of galls. The proper concentration for ACC and PZA is 500 mM and 250 mM, respectively.

### 3.3. Clubroot Induced ET-Response Genes during Infection

To explore the molecular mechanism of *A. thaliana* responses to *P*. *brassicae* with exogenous ethylene and inhibitor, gene expression of treated and untreated roots in different stages was detected at 7 dpi (the primary infection in root hairs and epidermal cells), 14 dpi (the secondary infection in cortical cells), and 20 dpi (the development of galls) by qRT-PCR. We focused on the gene expression by the treatment with 500 mM ACC and 250 mM PZA due to distinct clubroot symptoms (Figure 5). Firstly, the expression of *EIN3* and *EIL1* was detected in different stages. The expression level of *EIN3* and *EIL1* had almost no change at 7 dpi and 14 dpi (Figure 5A). Then, a number of transcription factors and genes related to ET and plant defense were tested. Previous studies have shown that both JA and SA contribute to resistance to clubroot [9]. *PR2*, an SA-responsive gene, was suppressed in infected *Col-0* but significantly induced by 500 mM ACC at 7 dpi and 20 dpi, while the expression level of *PR2* was induced in all treatments at 14 dpi (Figure 5B). However, *RGAH2* and *THI2*, two JA marker genes, were not induced by 500 mM ACC in all stages. In contrast, *RGAH2* was induced in infected *Col-0* at 14 dpi and 20 dpi, and *THI2* was induced 70-fold higher compared with *Col-0* at 20 dpi (Figure 5B). These results indicated that ET-regulated resistance was independent of JA, and SA might be induced by ET in response to clubroot disease at the early infection stage.

WRKY and ERF transcription factors are commonly found in the downstream of ET signaling. Although *WRKY45* was induced at 7 dpi and suppressed at 14 dpi by both clubroot and ACC, the effect of ACC was more significant compared with that of clubroot (Figure 5C). *WRKY75* was induced by 500 mM ACC at 7 dpi and suppressed at 14 dpi (Figure 5C). *AT2G20350*, a member of the B-6 subfamily of ERF/AP2 transcription factors, was intensively suppressed by about ten folds in infected *Col-0* at 7 dpi, but the relative expression was recovered by 500 mM ACC and suppressed by about two folds by 250 mM PZA compared with untreated *Col*-0 (Figure 5C). Another B-6 subfamily of ERF/AP2 transcription factor *SHN1* is involved in wax and cutin biosynthesis. It was found that *SHN1* was induced by 500 mM ACC and suppressed by 250 mM PZA at 7 dpi, and the expression was suppressed in all of the other treatments (Figure 5C).

*FER* was induced by 500 mM ACC and suppressed by 250 mM PZA at 7 dpi. The expression of RALFs and LRXs was detected, as shown in Figure 5D. At 7 dpi, RALFs were induced in infected *Col-0*, but they displayed higher expression by 500 mM ACC. The expression of *RALFL27* was decreased both in ACC and PZA treated roots, and the trend was more notable in PZA-treated roots. *RALFL24* showed the opposite expression pattern in the treatment between ACC and PZA. At 14 dpi, RALFs were upregulated in infected *Col-0*, and possessed similar expression patterns between the treatments with ACC and PZA. These results indicate that RALFs’ function in response to clubroot and ethylene may be involved in this process in the infection of clubroot. *LRX3* was induced by 500 mM ACC and suppressed by 250 mM PZA at 7 dpi, which shared the same expression pattern with *FER*.

### 3.4. Positive Effect of WRKY75 on the Infection of Clubroot in an ET-Dependent Manner

Since *WRKY75* can be induced by ethylene and SA, which is a direct target gene of *EIN3*, it is necessary to understand whether it participates in response to clubroot. The *wrky75-c* mutant, obtained by gene editing through the CRISPR/Cas9 system, was investigated via the inoculation test. A homozygous mutant with a G insertion at the 144th position of the coding region from the initiation codon was gained. *wrky75-c* displayed more susceptibility than *Col-0*, and *wrky75-c* showed delayed leaf senescence compared to *Col-0* (Figure 6), consistent with previously reported T-DNA insertion line N121525 (*wrky75-25*) [15]. However, the infection of *wrky75-c* revealed early-senescence in leaves compared to both *Col-0* and untreated *wrky75-c*. Of note, the clubroot symptoms in *wrky75-c* were similar to the mutants, such as *ein3/eil1*, *ein2*, and *etr1-1* [29], which represented extremely short roots and barely lateral roots. It is speculated that *WRKY75* may have a positive role in response to clubroot through the ethylene pathway during the early infection.

## 4. Discussion

Previous studies have shown that ET is a positive regulator against *P. brassicae*. A number of mutants related to ET, such as *ein2*, *ein3-1*, *ein4*, and *etr1-1*, have been reported to be more susceptible than the wild-type in *Arabidopsis* [29]. In our study, *ein3/eil1* double mutant and *ein3* and *eil1* single mutants were tested in response to clubroot. *ein3/eil1* and *eil1* represented increased clubroot symptoms compared to *Col-0* (Figure 1), which is consistent with previous studies and provides further proof that ET is needed against *P*. *brassicae.* However, the clubroot symptoms of *ein3* were opposite from previous studies. Several studies have shown that some phytohormone-impaired/insensitive lines in *Arabidopsis* have a positive or negative effect on different root knot nematodes that may be due to various root exudates [30]. For example, the ET-regulated pathway reduces the root attractiveness of the soybean cyst nematode but increases the attractiveness of the sugar beet cyst nematode *Heterodera schachtii*. We assume that the differences between the isolation e3 and P4 may be the cause of clubroot symptoms of *ein3*. Meanwhile, the transcription level of *PR2* and *THI2* was different in infected *Col-0* compared to untreated *Col-0* (Figure 5B) [9], which may contribute to the special properties between eH (belongs to P1) and P4 of *P. brassicae*. In addition, recent studies have shown that root galls caused by *P. brassicae* are a mixture of multiple strains [31]. The pathogens of *P. brassicae* in our study (P4) were obtained from the field, although as identified as P4 by Professor Zhongyun Piao, they may contain more than one strain, while e3 and eH are both single strains. Taken together, our results indicate that the mechanism of clubroot resistance may be different among different pathotypes of *P. brassicae*.

Although ethylene has a positive role in resistance to clubroot in *Cruciferae*, detailed studies have yet to be conducted. The effect of ethylene on clubroot is mostly focused on the transcriptional level between clubroot-resistant and clubroot-susceptible roots. Inoculation of 5 × 10^5^ spores at mL^−1^ for the ethylene signaling pathway and biosynthesis mutants leads to more susceptibility than *Col-0*, displaying almost no clubroot symptoms [29]. However, *eto-2*, an ET-overproducing mutant, also showed more susceptibility compared with *Col-0*. Yet the clubroot symptoms of *eto-2* were reduced compared with *Col-0* and ET signaling pathway and biosynthesis mutants when higher concentrations of spores (1 × 10^6^ mL^−1^ to 1 × 10^7^ mL^−1^) were used. All of these results revealed that a sophisticated process was governed by ET in response to clubroot. The application of the exogenous ethylene precursor ACC and ACC oxidase inhibitor PZA changed the clubroot symptoms under the 1 × 10^7^ mL^−1^ pathogen. The clubroot symptoms between 20 dpi and 30 dpi showed a remarkable difference, and the clubroot symptoms treated with 500 mM ACC were reduced at 20 dpi but increased at 30 dpi compared with those treated with 250 mM PZA. Although the galls of roots treated with 500 mM ACC were bigger than those by 250 mM PZA at 30 dpi, the roots by 250 mM PZA had rot symptoms (Figure 4 and Figure 5). Therefore, it is hypothesized that ET has a dual role in response to clubroot by delaying the establishment of infection but promoting the development of galls.

Plant hormones are thought to play a vital role in plant response to various pathogens. Thus far, the defense mechanisms in leaves have been studied adequately, but the knowledge about root response to pathogens remains elusive. JA and SA, two antagonistic phytohormones in the classical plant immune systems in leaves, have been reported to have a positive effect on resistance to *P*. *brassicae*. However, the regulation process is still elusive. SA is thought to have a positive role in clubroot resistance because the *cpr5-2* mutant (which constitutively promotes the SA signaling pathway is more resistant to clubroot disease than the wild type [9]. However, the *ein3/eil1* double mutant, which constitutively accumulates SA, is more susceptible compared with the wild type [32]. It seems that the signaling pathway mediated by SA is blocked by *P*. *brassicae*, which has been confirmed with the discovery of methyltransferase (*PbBSMT*) secreted by *P*. *brassicae*. *PbBSMT* can inactivate SA by converting SA to methyl salicylate (MeSA) [33]. Herein, the SA signaling pathway was promoted by exogenous ACC treatment because *PR2*, a member of the SA signaling pathway, was significantly induced by 500 mM ACC compared with the wild type and the treatment by 250 mM PZA at 7 dpi. Moreover, the robust resistance to *P. brassicae* in the *bik1* mutant may also be related to SA accumulation due to the high *PR1* levels in *bik1* compared with the wild type [34]. ET is usually thought to act synergistically with JA in the defense response against necrotrophic pathogens, but interestingly, two JA marker genes, *THI2* and *ARGAH2,* were not induced by 500 mM ACC at 7 dpi and 14 dpi. Instead, *THI2* and *ARGAH2* were significantly induced in the infected *Col-0* at 14 dpi and 20 dpi compared to the untreated *Col-0* (Figure 5B). Although the expression levels of *THI2* and *ARGAH2* were induced by 500 mM ACC compared with *Col-0* at 20 dpi, the relative expression was lower in contrast to the infected *Col-0* and the treatment with 250 mM PZA. *ARGAH2* can enhance the expression of *NATA1,* thereby decreasing the clubroot symptoms [9]. Based on these results, we conclude that ET may contribute to clubroot resistance through activating the SA signaling pathway during early infection, but may promote greater susceptibility through inhibiting the JA signaling pathway during the second infection of clubroot.

Based on the above results, we investigated the regulation mechanism of ET response to clubroot during the early infection. *WRKY75*, *WRKY45,* and *SHN1* shared similar expression patterns, which were significantly induced by 500 mM ACC compared with the others (Figure 5C). Although *AT2G20350* was not significantly induced by 500 mM ACC, the expression was about ten times lower in the infected *Col-0* and by the treatment with 250 mM PZA compared with the untreated *Col-0* at 7 dpi. It indicates that these transcription factors may affect the resistance to clubroot, which was partially supported by the results of the inoculation of P4 for the mutant *wrky75-c*. It has been reported that *WRKY75* is ethylene-induced and directly targeted by *EIN3* [15]. *WRKY75* can promote SA accumulation by inducing SA synthesis gene *SID2* during leaf senescence [15], and *WRKY75* participates in a variety of other biological processes such as phosphorus stress, root hair patterning, and flowering. Likewise, *WRKY45* is involved in leaf senescence and phosphorus stress, which is directly regulated by *WRKY75* during leaf senescence [35,36,37]. *WRKY45* may be mediated by *WRKY75* during the ET treatments at the early infection of clubroot, while the relationship between *WRKY45* and SA remains unknown. Both *SHN1* and *AT2G20350* belong to B-6 of the ERF/AP2 transcription factor family, and most of their functions remain unclear. *SHN1* is involved in positively regulating defense to *Botrytis cinerea* [38,39]. It is reported that the ethylene signaling pathway can limit the pathogen to outer cell layers. For example, the ethylene pathway can prevent the invasion of oomycete *Pythium irregular* into the epidermis and outer and inner cortex of the tap roots [40]. *SHN1* regulates the contents of the cuticle, which consist of wax, and is the first barrier to prevent plants from invading pathogens. *SHN1* may contribute to the delaying of infection during the early infection. However, studies on *AT2G20350* have not been documented to date, and hence the functions remain unknown.

Membrane-localized receptors have vital roles in response to distinguishing nonbeneficial components during the infection of pathogens [41]. *FER*, a sensor of cell wall integrity, together with RALFs and LRXs, regulates the immune signaling pathway during host–pathogen interactions [42]. It is reported that the function of *FER* in plant defense differs due to the lifestyle of pathogens [43]. In our study, the RALFs–LRXs–FER pathway was induced by 500 mM ACC at 7 dpi, which may have a positive effect on clubroot invasion. In addition, *WRKY75* and *WRKY45* can be upregulated in *FER* mutant leaves [43]. *P. brassicae* prefers acidic soils with a pH value of less than 5.5, so it may hinder the uptake of Ca, K, Mg, and P [44]. Taken together, the RALFs–LRXs–FER signaling pathway induced by ET may function through two pathways, i.e., by regulating *WRKY75* and *WRKY45* and by mediating the soil alkalization.

## 5. Conclusions

Our study found that ET was required in the plant–*P. brassicae* interaction, and ET had a dual role in the infection of *P. brassicae* in *A. thaliana*. ET activated the SA signaling pathway and a number of transcription factors including *WRKY75*, *WRKY45*, *SHN1,* and *At2G20350*, hence delaying the establishment of infection during the primary infection. In contrast, the JA pathway was suppressed, thereby promoting the development of galls during the secondary infection.

## Figures and Tables

**Figure 1 genes-13-01299-f001:**
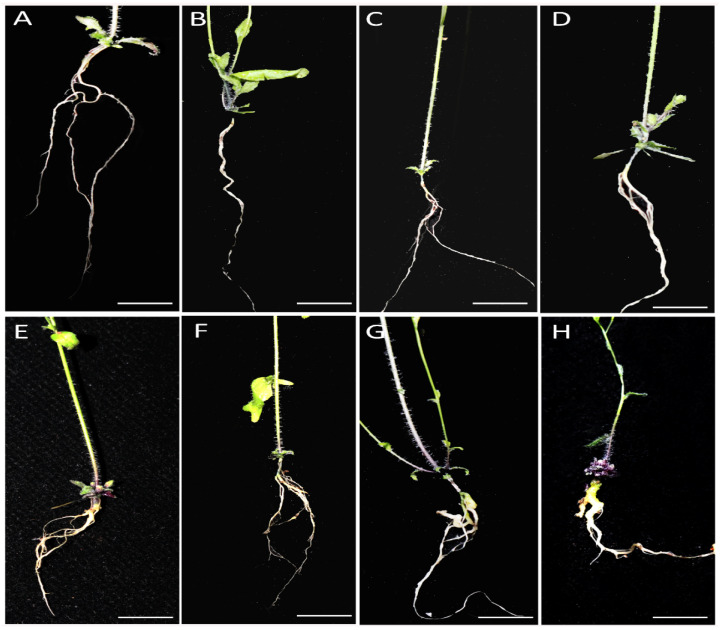
Phenotypes of infected and uninfected roots of wild type and mutant lines after 30 days of inoculation. (**A**–**D**) were control lines while (**E**–**H**) were infected lines with the same order: *Col-0*, *ein3*, *eil1*, and *ein3/eil1*.

**Figure 2 genes-13-01299-f002:**
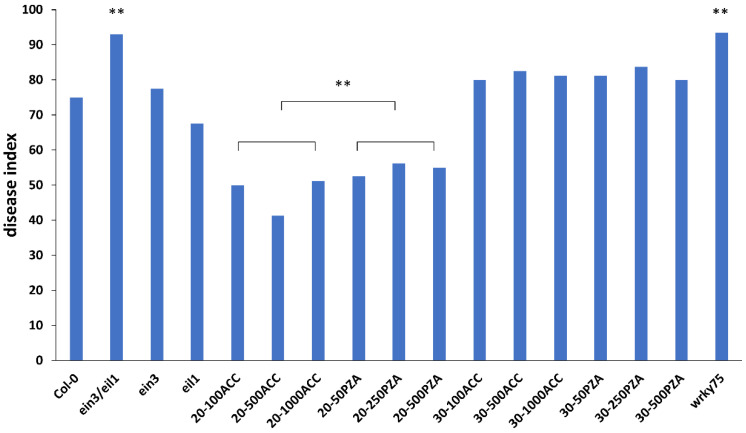
Disease index of different materials and treatments. Col-0, ein3/eil1, ein3, eil1, and Col-0 treated with 100 mM ACC, 500 mM ACC, 1000 mM ACC, 50 mM PZA, 250 mM PZA, and 500 mM PZA at 20dai and 30dai lines were evaluated with the 0–4 scoring system. For each line, 40 plants were analyzed. Student’s *t* test, ** *p* < 0.01.

**Figure 3 genes-13-01299-f003:**
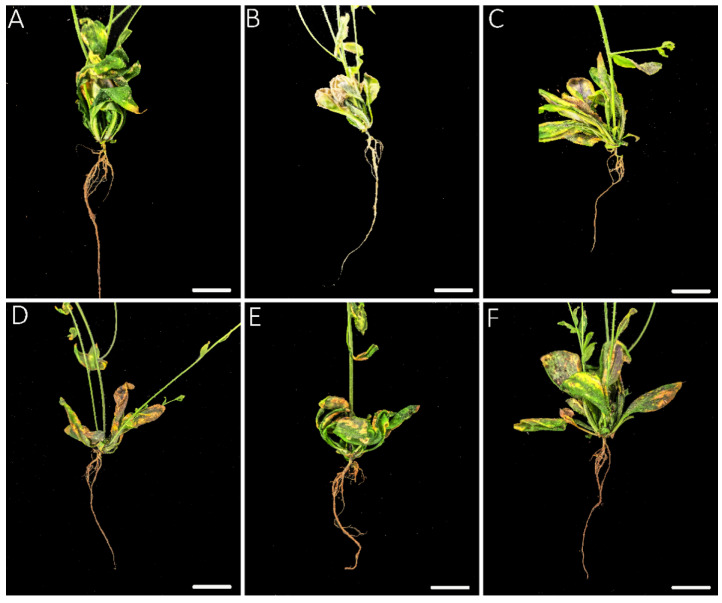
Clubroot symptoms of infected roots with exogenous ACC and PZA in *Arabidopsis*. (**A**): 100 mM ACC; (**B**): 500 mM ACC; (**C**): 1000 mM ACC; (**D**): 50 mM PZA; (**E**): 250 mM PZA; (**F**): 500 mM PZA. Roots were photographed at 20 dpi.

**Figure 4 genes-13-01299-f004:**
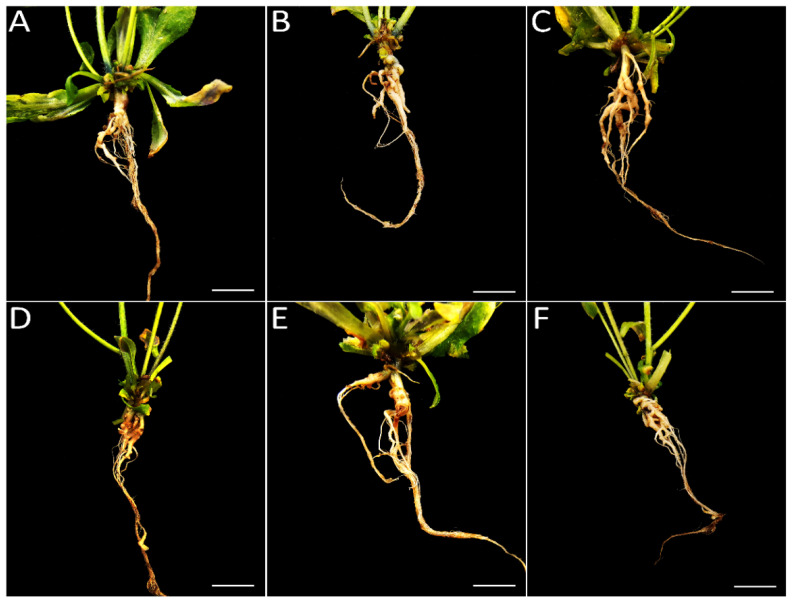
Clubroot symptoms of infected roots with exogenous ACC and PZA in *Arabidopsis*. (**A**): 100 mM ACC; (**B**): 500 mM ACC; (**C**): 1000 mM ACC; (**D**): 50 mM PZA; (**E**): 250 mM PZA; (**F**): 500 mM PZA. Roots were photographed at 30 dpi.

**Figure 5 genes-13-01299-f005:**
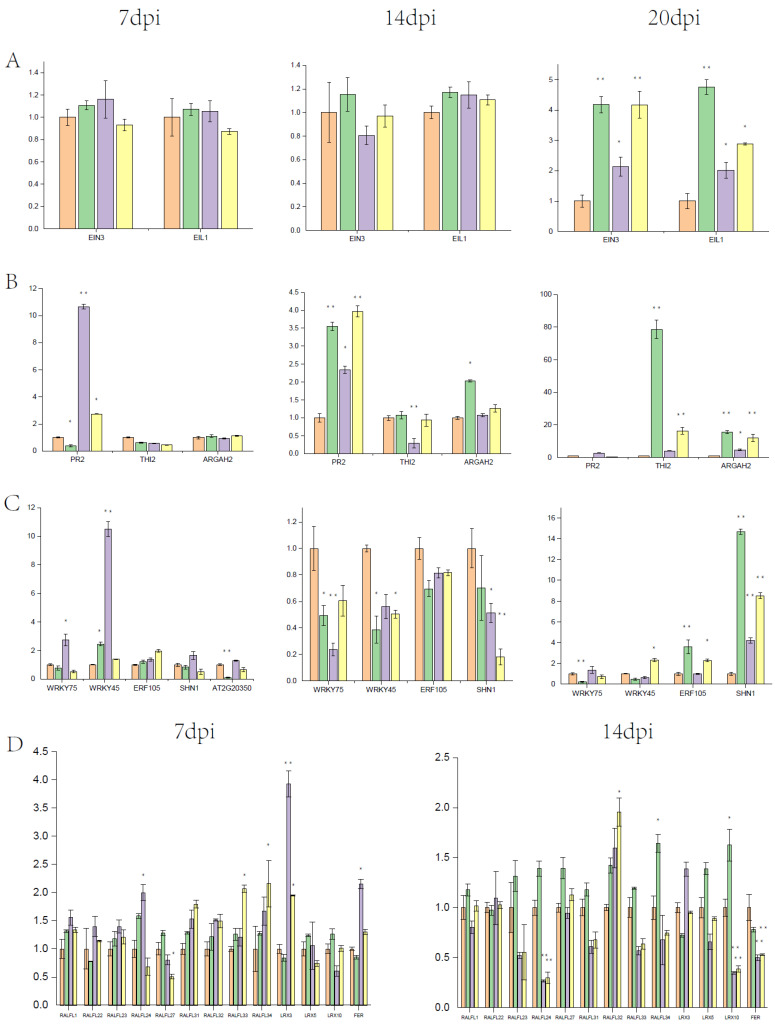
Relative expression of ET- and clubroot-related genes at 7 dpi, 14 dpi, and 20 dpi, respectively. (**A**): The relative expression of *EIN3* and *EIL1*; (**B**): The relative expression of SA and JA marker genes; (**C**): The relative expression of ET-regulated transcription factors; (**D**): The relative expression of genes in the RALFs–LRXs–FER pathway. The samples were untreated *Col-0*, treated *Col-0*, treated *Col-0*-500 mM ACC, and treated *Col-0*-250 mM PZA from the left. Data are means (±SD) (*n* = 3). Student’s *t*-test, * *p* < 0.05, ** *p* < 0.01.

**Figure 6 genes-13-01299-f006:**
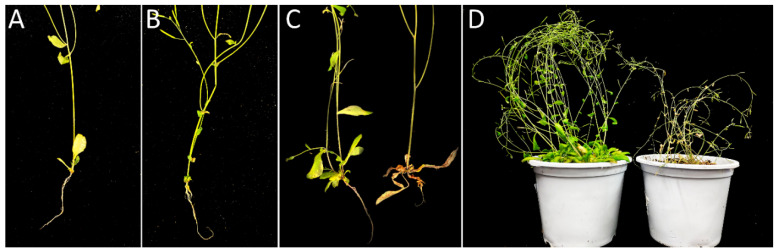
Phenotype of infected and uninfected roots in wild type and *wrky75-c* after 30 days of inoculation. Inoculation *P*. *brassicae* of two-week-old seedlings. Roots were photographed at 30 dpi. (**A**): Control, (**B**): infected *Col-0*, (**C**,**D**): starting from the left: uninfected and infected *wrky75-c*.

## Data Availability

All data generated or analyzed during this study are included in this published article (and its Appendix A).

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
