# Peer review of "Ethylene Plays a Dual Role during Infection by Plasmodiophora brassicae of Arabidopsis thaliana"

_genes, 2022, doi:10.3390/genes13081299_

Round 1

Reviewer 1 Report

The article studies the response of A. thaliana to P. brassicae infection. The putative role of ethylene was studied in two single (ein3, eil1) and one double mutant ein3/eil1. Moreover the ethylene precursor (ACC) and the ACC oxidase inhibitor (pyrazinamide) were applied into the study.

The study is generally well designed and performed, results support the conclusions. In my opinion the methods section should be better described to  assure that the results are reproducible. Also there is many typographical errors that should be corrected before the publication.

Line 21; do not capitalize W, in the We

Line 84 ; remove brackett after 2012

Line 94- P. brassicae should be in italics, check also line 106 and the entire text

Line 96, 296; pyrazinamide is not an inhibitor of ACC but the ACC oxidase, correct the sentence.

Line 124; EcoRI remove space between R an I.

Line 159; lack of space after root.

Line 187; remove rot.

Line 274, 341, 351; remove the additional brackett.

Line 362; citation should be Herger et al. , 2019. Check the citation format also in line 34 and the entire text.

Section 2.3

The methodology used to assure RNA quality.

Amount of RNA or cDNA used for analysis.

Details of PCR cycling reaction.

Method used to calculate RT-PCR results with citation. 

Statistical methods used to analyse results.

Section 2.4

Plantlets were soaked in the spore solution or spread by this solution?

Line 142/243; describe more clearly the application of ACC and PZA, it is not clear how it was added in the clubroot fluid.

There is no description of statistical methods applied to other parts of manuscript- not only RT-PCR. Add description of statistical methods to other quantitative parts of research experiments as disease index calculation.

Figure 5; on the figure is written dai and in the figure description dpi; use the same on both.

Reviewer 2 Report

Comment: Minor revision

This is an interesting study focused on host-pathogen interaction. This study sheds some more light on the role of ET in Plasmodiophora brassicae infection in Arabidopsis thaliana. This study would help in more detailed study of the function of ET in plant root defense against pathogens in the future. However, the strength of the manuscript is greatly affected by some important issues pointed out below. The author needs to be careful with grammatical mistakes and readability. The author should revise the manuscript more carefully and scientifically.

Line 16-17: Rewrite the sentence. The word “Some researcher” needs to be replaced with a better confident word.

Line 21-23: Replace “We” with “we”. Write the abbreviation Col-0 as it is mentioned here for the first time. Remove the concentrations of the ACC and PZA from the abstract. This concentration should be mentioned in the methods section.

Line 29: Write the organism’s scientific names and gene names in italic font.

Line 39-40: Rewrite the sentence clearly.

Line 47-51: Breakdown the sentence and write clearly.

Line 63: Add the word “pathways” after the word “signaling”.

Line 104: Replace the word “edit” with “editing”.

Line 142-143: Write the concentration of the ACC and PZA clearly so that the concentrations can easily be distinguishable.

Line 173 and 210: You stated “Previous studies have shown that…..” but you did not provide appropriate citations of the previous study related to the topic you discussed. For citation follow the guideline of the journal.

Figure 5: The color contrast of the graphs is not good. Some graphs are difficult to understand. Use good quality figures.
